# Food Purchasing Characteristics and Perceptions of Neighborhood Food Environment of South Africans Living in Low-, Middle- and High-Socioeconomic Neighborhoods

**Feyisayo Odunitan-Wayas [1], Kufre Okop [1,2], Robert Dover [3], Olufunke Alaba [4], Lisa Micklesfield [1,5], Thandi Puoane [2], Monica Uys [1], Lungiswa Tsolekile [2], Naomi Levitt [6], Jane Battersby [7], Hendriena Victor [1], Shelly Meltzer [8] and Estelle V. Lambert [1,***

[1]  University of Cape Town Research, Centre for Health through Physical Activity, Lifestyle and Sport, Division of Exercise Science and Sports Medicine, Department of Human Biology, Faculty of Health Sciences, University of Cape Town, Cape Town 7725, South Africa; feyi.odunitan-wayas@uct.ac.za (F.O.-W.); kufre.okop@uct.ac.za (K.O.); Lisa.Micklesfield@uct.ac.za (L.M.); monika.uys@hotmail.com (M.U.); hendriena.victor@uct.ac.za (H.V.)

[2]  School of Public Health, University of the Western Cape, Bellville, Cape Town 7535, Western Province, South Africa; tpuoane@uwc.ac.za (T.P.); ltsolekile@uwc.ac.za (L.T.)

[3]  Departamento de Antropología, Universidad de Antioquia, Medellin 050010, Colombia; rvhdover@gmail.com

[4]  Health Economics Division, School of Public Health and Family Medicine, Faculty of Health Sciences, University of Cape Town, Cape Town 7925, South Africa; olufunke.alaba@uct.ac.za

[5]  MRC/Wits Developmental Pathways for Health Research Unit, Faculty of Health Sciences, University of the Witwatersrand, Johannesburg 2000, South Africa

[6]  Department of Medicine, Faculty of Health Sciences, University of Cape Town, Cape Town 7925, South Africa; Naomi.Levitt@uct.ac.za

[7]  African Centre for Cities, University of Cape Town, Cape Town 7701, Western Province, South Africa; jane.battersby.lennard@gmail.com

[8]  Shelly Meltzer and Associates, Nutrition and Dietetics, Sports Science Institute of South Africa, Newlands, Cape Town 7700, South Africa; meltzer@iafrica.com

*  Correspondence: vicki.lambert@uct.ac.za; Tel.: +27-823-126-890

**Abstract:** Using intercept surveys, we explored demographic and socioeconomic factors associated with food purchasing characteristics of supermarket shoppers and the perceptions of their neighborhood food environment in urban Cape Town. Shoppers (N = 422) aged $\geq$18 years, categorized by their residential socioeconomic areas (SEAs), participated in a survey after shopping in supermarkets located in different SEAs. A subpopulation, out-shoppers (persons shopping outside their residential SEA), and in-shoppers (persons residing and shopping in the same residential area) were also explored. Fruits and vegetables (F&V) were more likely to be perceived to be of poor quality and healthy food not too expensive by shoppers from low- (OR = 6.36, 95% CI = 2.69, 15.03, $p < 0.0001$), middle-SEAs (OR = 3.42, 95% CI = 1.45, 8.04, $p < 0.001$) compared to the high-SEA shoppers. Low SEA shoppers bought F&V less frequently than high- and middle-SEA shoppers. Purchase of sugar-sweetened beverages (SSBs) and snacks were frequent and similar across SEAs. Food quality was important to out-shoppers who were less likely to walk to shop, more likely to be employed and perceived the quality of F&V in their neighborhood to be poor. Food purchasing characteristics are influenced by SEAs, with lack of mobility and food choice key issues for low-SEA shoppers.

**Keywords:** shopping behaviors; food environment; food insecurity; food purchasing characteristics; socioeconomic area; obesity; out-shoppers

## 1. Introduction

Obesity and food insecurity (i.e., lack of sufficient physical and economic access to enough quality and nutritious food for an active and healthy life) [1], often co-exist in South Africa, and are associated with poor diet and health outcomes [1]. Food choice behaviors are considered to be largely influenced by the food environment and socioeconomic status [1]. The retail food environment, which is defined as food outlets within a person's neighborhood [2] has been linked to diet quality [3]. Socioeconomic status of individuals has also been associated with diet quality in several studies in both developed and developing countries, although with varying predictors of socioeconomic status, such as education, occupation, income and area of residence [4–6]. However, there is limited evidence from lower- and middle-income countries and the Global South.

Only a small number of studies have examined informal food retail outlets, such as 'spaza shops' (an informal convenience shop in residential neighborhoods [7]) and street vendors in relation to shopping decisions and dietary choices. However, although more is known about the geospatial distribution of supermarkets in South Africa [8–11], there is limited evidence on the association between supermarket locations and food purchasing characteristics in low-, middle- and high-socioeconomic areas. In South Africa, formal retailers, such as supermarkets and fast food outlets and less formal retailers, such as street vendors, convenience stores, and in certain areas, food/community markets are the predominant source of food shopping [12]. There has however been a rapid expansion of supermarkets over the last decades and supermarkets have become a primary source for food shopping, accounting for more than 50% of food sales in South Africa [12,13]. Furthermore, most people in urban areas purchase their food items from supermarkets [14–16]. For instance, more than 90% of the population in Cape Town shop in supermarkets [17,18]. As such, supermarkets have been identified as a potential role player in curbing obesity and addressing food insecurity, by providing local access to healthy food, especially in urban communities [19,20].

Nevertheless, several studies have shown that access to supermarkets does not necessarily lead to healthy food choices, as they increase access to both healthy and unhealthy food choices [21,22]. Also, it has been documented that the expansion of supermarkets in South Africa is not evenly distributed, with a higher concentration in middle and high income areas [19]. Furthermore, the expansion of supermarkets in South Africa has been largely characterized by a movement of the more discount retail chains into low income areas, and associated foods of lower quality [19]. The results are a promulgation of "food deserts," when it becomes challenging to buy affordable and/or quality healthy foods and consequently, persons from these communities may lack access and sufficient economic resources to purchase healthy food [19]. In some cases, residents of an area may choose to purchase food in supermarkets outside of their neighborhood (out-shopping) as they are often dissatisfied with the quality or variety of foods in their local environment and may drive or travel a long distance for better access to healthy foods or better value at discount supermarkets [10,23]. Furthermore, car ownership, high level of education and higher socioeconomic status have been associated with outshopping in South Africa [10]. The underlying factors associated with "out-shopping" are varied and may be related to individual and demographic factors, or more broadly, related to the local food environment. This has not been widely studied in lower- and middle-income country settings [10,14].

Characterizing food purchasing decisions, as well as perceptions of the neighborhood retail food environments, in shoppers from different socioeconomic areas, may add to our understanding of factors that influence food choice behaviors, and that may impact, directly or indirectly on health. The overall goal of the present study is to advance our knowledge concerning those factors that leverage or impact on the food purchasing decisions of shoppers residing in low-, middle- and higher socioeconomic areas in urban suburbs of Cape Town, South Africa. The aims of the study are to better understand the shopping characteristics of supermarket shoppers from low, middle and high

socioeconomic areas; to explore the perceptions of the low, middle and high socioeconomic area shoppers of their neighborhood food environment; and to unpack the food purchasing characteristics and neighborhood food environment perceptions of out-shoppers vs. in-shoppers.

## 2. Material and Methods

### 2.1. Study Design and Settings

The current cross-sectional analytical study is part of a larger study, STOP SA (Slow, Stop or Stem the Tide of Obesity in the People of South Africa), aimed at addressing the challenges of obesity in conjunction with food insecurity. Data for this study were collected between March and May 2017 in 11 major supermarkets located in purposively selected two high-, two middle- and two low-SEAs in Cape Town.

### 2.2. Socioeconomic Profile of Cape Town

The selected areas are based in Cape Town, the provincial capital of Western Cape, and second most populous city in South Africa with a population size of approximately four million people [24]. The selected areas were categorized into three categories (low, middle and high socioeconomic areas) based on the 2016 Socio-economic Profile: City of Cape Town [25] which was classified using the average household income.

In the current study, the selected low socioeconomic areas were Langa and Khayelitsha. Both areas are townships located in the Cape Flats, which is one of the poorest sections in Cape Town. The Langa population of approximately 52,500 has an average monthly household income of ZAR2144 [26]. Khayelitsha, the fastest growing township in Cape Town with a population of approximately one million people, has the highest poverty rate in Cape Town [17], with more than 60% of the population having an average monthly household income of ZAR1600 [27]. The townships areas (Khayelitsha and Langa) cover 38.71 and 3.09 square kilometers respectively [25]. The middle SEAs selected were Athlone with a population size of 237,000 and Mowbray with a population size of 5000. Both middle SEAs have an average monthly household income of ZAR5217 [28]. The high SEAs included Parklands, one of the fastest growing new residential areas covering 2.47 km$^2$ with a population size of about 43,000 and average monthly income of ZAR9500, and Claremont, an old residential area covering 5.21 square kilometers with a population size of 17,000 and average monthly income of ZAR12000 [24]. Based on the 2016 Socio-economic Profile: City of Cape Town [25], shoppers and the supermarkets were classified into low, middle and high socioeconomic categories by residential area and location respectively.

### 2.3. Supermarket Sample

Supermarkets in this study are major recognized retail store chains in South Africa that offer a broad selection of foods and household products. The managers of major supermarkets in the selected study areas were approached to obtain permission to conduct our study within their premises, but outside the supermarkets. Consent from supermarket managers was obtained from 11 supermarkets (four supermarkets in low SEAs, five in middle SEAs and only two supermarkets in the high SEAs) representing approximately 20% of the total supermarkets in these areas. The 11 supermarkets represented three of the four major recognized retail store chains in South Africa. We were unable to obtain consent from many of the managers of supermarkets in high SEAs as they did not wish their customers to be disturbed.

### 2.4. Intercept Survey

We intercepted shoppers that were coming out of pre-selected supermarkets and invited them to answer a short structured survey which was interview administered. Intercept surveys involve stopping target group (shoppers), screening them for eligibility of the study and administering a survey on the spot which is usually in a public or business place. The intercept survey was piloted

in a preliminary test in two supermarkets using similar methods described in the current study. Each shopper was approached after shopping and briefed on the objective of the study, and after voluntarily consenting to participate, and signing an informed consent, he/she was interviewed. Only shoppers who were ≥18 years old and had purchased more than 10 different items confirmed by their grocery receipts were eligible for the intercept survey. A total of 635 shoppers were approached, of which, 425 agreed to be part of the study. The remainder, of which more than 60% were from the high SEAs, cited time constraints as the major reason for non-participation. Overall, information on only 422 shoppers from the 11 supermarkets was used as three interviews were incomplete. Each intercept survey lasted 20–25 min and was conducted by a trained research assistant/fieldworker in either one, or a combination of the three major languages in Cape Town: English, Xhosa and Afrikaans, depending on the preference of the shopper. The intercept surveys were conducted between 10:00–17:00 on weekdays, and 10:00–14:00 on Saturdays in the beginning, middle and end of the month in each of the supermarkets to capture various categories of shoppers. The structured questionnaire included information concerning shopping patterns and practices, perceptions of the neighbourhood food environment, demographic characteristics and the food security status of the shopper. Participants were given a shopping voucher (ZAR50/≈$4) as compensation for their time after completing the intercept survey.

*2.5. Out-Shoppers and In-Shoppers*

Previous research has indicated that sometimes for various reasons, some people tend to shop outside their residential areas. They also indicated that these shoppers may have certain common factors or drivers that results in their out-shopping [10,23]. In the present study, we also looked at a sub-population based on their shopping socioeconomic area versus their residential socioeconomic area. This sub-population was classified into two groups; out-shoppers and in-shoppers. The proportion of shoppers shopping within their residential SEA are defined as in-shoppers and those shopping outside their residential SEA as out-shoppers. The aim is to explore this subpopulation to better understand the characteristics of in-shoppers and out-shoppers.

*2.6. Measures*

The 35-item intercept survey had six main sections:

2.6.1. Shopping Characteristics and Mode of Transportation to the Supermarket

This section of the questionnaire consisted of questions on self-reported frequency of shopping (daily, weekly, monthly), number of persons shopped for, the main person in the household responsible for shopping, main person in the household responsible for food preparation, to indicate by ranking the most important factors that influence their choice of supermarket (e.g., price, convenience, proximity, quality, value for money and variety), the major shopping place(whether the supermarket in which they were intercepted was their main shopping place), and the mode of transportation to supermarket.

2.6.2. Food Types and Frequency of Purchase

Questions in this section included the self-reported frequency of purchase of various foods, such as meat (fresh, frozen, dried, whole, portioned); fruits and vegetables (fresh and frozen), and snacks (chips, sweets, chocolates and cakes).

2.6.3. Self-Report on Bread and Sugar-Sweetened Beverages (SSBs)

Information on bread type preference and self-reported purchase of sugar-sweetened beverages (soft drinks/sodas, flavored juice drinks, non-alcoholic wine, flavored water with sugar, sports drinks, energy drinks and fruit juice blends, cordials, fruit nectar and all fruit juices) were elicited. Bread is generally purchased by the average South African [3], however, the knowledge gap is the bread

type preference, which is important in this paper as the focus is on healthy food choice, affordability and availability.

The question about the purchase of SSBs varied from the other categories as the consumption of SSBs has increased significantly over the years and it was often something that most shoppers purchase each time they shop [11]. Asking if they purchased SSBs will be more significant in the representative consumption of SSBs as opposed to asking how frequently they purchased SSBs.

### 2.6.4. Neighborhood Food Environment Perceptions

We measured the perceptions of shoppers' neighborhood food retail environment using adapted four statements from a previous study conducted in a low income neighborhood in the United States [29]. Each statement was designed to address a distinct dimension of the retail neighborhood food environment. One pertained to the general neighborhood retail food environment, two were specific to fruits and vegetables in the neighborhood and one to the affordability of healthy foods in their neighborhood (formal and less formal retail outlets). Representative statements included 'There are no supermarkets in my neighborhood', 'It is easy to purchase fruits and vegetables in my neighborhood', 'The healthy foods in stores in my neighborhood are too expensive' and 'The quality of fruits and vegetables in my neighborhood is poor'. Shoppers were asked these statements using a 5-item Likert scale coded 1–5 (1 strongly agree, 2 agree, 3 neither agree or disagree (neutral), 4 disagree, 5 strongly disagree).

### 2.6.5. Demographic Characteristics

Questions in this section of the intercept survey included self-reported sex, age in years, residential location and three indicators of an individual's socioeconomic position, specifically: Educational attainment, employment status and socioeconomic area. Age was categorized into three groups: Young adults early working age (18–30), prime working age (31–55), and mature working age and seniors (>55 years). Educational attainment and employment status were both self-reported. Educational level was grouped into three categories: Primary, high school, and tertiary education. Employment status was classified as employed, unemployed, homemaker or retired. Socioeconomic area was determined by categorizing the shopper's self-reported residential location according to the socioeconomic profile of the city of Cape Town 2016 [24].

### 2.6.6. Food Security Assessment

Three key food security questions were adapted from the U.S. Household Food-Security/Hunger Survey Module: 3-Stage Design [17]. The questions were: (i) In the last 12 months, did you eat less than you felt you should because there wasn't enough money for food? (ii) In the last 12 months, were you hungry but didn't have enough money for food? (iii) How often were you able to eat a balanced diet in the last 12 months? An affirmation to questions i and ii and "often" and "sometimes" to question iii were coded "yes" for food insecure. A negative response to question i and ii and "never true" to question iii were coded "no" for food insecure to create a binary variable for food insecurity (0 = insecure and 1 = secure).

### 2.7. Data Analysis

Descriptive analyses were undertaken. Pearson chi-square analysis was used to determine significant differences in demographics, shopping characteristics and purchase frequency of food categories between residential SEAs, and between out-shoppers and in-shoppers within the low and middle SEA groups. The Likert scale (1–5) for measuring neighborhood food environment perceptions were subsequently collapsed into three categories and recoded as: Strongly agree and agree = 1 (agree); somewhat agree or disagree = 2 (neutral); disagree and strongly disagree = 3 (disagree). Thereafter, multinomial regression analyses were conducted on the dependent variables (perceptions of the neighborhood environment) to understand the associations between individual-level factors (education

level, employment, residential area, food security status) as independent variables. Food environment perceptions were controlled for individual-level effects, such as age, sex, education and employment. Multinomial regression is often used to predict the nominal dependent variable with more than two categories for one or more independent variables. We tested for multicollinearity between the independent variables and found no substantial issues as no variation inflation factor (VIF) exceeded 3. The association of shopping characteristics and neighborhood food environment perceptions as independent variables and out-shopping as the dependent variable was also analyzed using logistic regression controlling for age and sex. Data were analyzed using IBM SPSS for Windows, version 24, Armonk, New York: IBM Corporation

## 3. Results

### 3.1. Demographic and Shopping Characteristics

The shoppers' demographics and shopping characteristics by shoppers' residential socioeconomic status are presented in Table 1. There were significant differences in age distribution, education level, employment and modes of transport between persons from the three SEAs. More than half (60%) of the shoppers were between the ages of 30–55 years and 82.5% (N= 344) were women. Just over half of the participants from the low SEAs had a primary school education while a similar proportion of participants from the high SEAs had tertiary education. Unemployment was more common in the participants from low SEAs than in shoppers from high SEAs (36.3% vs. 13.0%). Most respondents from the low SEAs walked to shop (67.2%) and more than half (62.1%) spend more than 10 min to get to the supermarkets from their home, while most participants from the high SEAs used a private car (73.2%) and spent less than 10 min or less (88.7%) to get to the supermarket. According to the food security assessment, food insecurity status was inversely associated with SEAs ($p < 0.001$).

Of the people interviewed, most were the household member primarily responsible for grocery shopping (78.9%) and food preparation (79.6%). When compared to residents from high SEAs, a higher proportion of residents from low and middle SEAs were mostly responsible for major grocery shopping (81.7% and 83.3% vs. 62%, $p = 0.01$). Similarly, more persons from the low and middle SEAs than from high SEAs were responsible for food preparation in their households (79% and 89.6% vs. 70.4%, $p = 004$). For most (85.1%) of shoppers, the supermarket in which they were interviewed was the one in which they mostly shopped. Sales/promotions were not indicated as an important factor for choice of supermarket in any of the SEAs. The highest proportion of the respondents (46.7%), irrespective of their residential SEAs were weekly shoppers compared to daily (28.4%) and monthly (24.9%) shoppers, with most (74.4%) purchasing specifically for their own households. Price and convenience were two factors that were most frequently indicated as important in the choice of supermarkets, however the most common factor differed significantly between the groups ($p < 0.001$), with price being the most common selected factor of shoppers from the low and middle SEAs, and convenience in the high-SEA group (Table 1). None of the high-SEA residents were classified as out-shopper (persons shopping outside their residential socioeconomic area) compared to 12.0% from middle SEAs and 23.7% from low SEAs. Therefore, in this study, we will be exploring more on the characteristics of out-shoppers and in-shoppers from only the low and middle socioeconomic areas for further comparative analysis.

### 3.2. Purchase Frequencies and Preferences

Self-reported purchase frequencies of food categories according to the respondent's residential SEAs are presented in Table 2. Persons living in low SEAs were likely to purchase fruits and vegetables and meat less frequently than persons from high and middle socioeconomic areas, but there was no difference in the frequency of purchasing snacks and SSBs between the shoppers from different SEAs. Brown bread was the most preferred bread type by all the shoppers. However, more low-SEA shoppers (69.6%) preferred brown bread compared to 40.5% middle-SEA and 52.1% high-SEA shoppers ($p < 0.001$).

**Table 1.** Demographic and shopping characteristics of participants by residential socio-economic areas.

| Variables | Residential SEAs | | | | |
|---|---|---|---|---|---|
| | High *SEAs* n (%) | Middle SEAs, *n* (%) | Low SEAs, *n* (%) | Total | *p*-Value |
| N (%) | 71 (16.8) (%) | 132 (31.3) (%) | 219(51.9) (%) | | |
| **Demographics** | (%) | (%) | (%) | (%) | |
| Age | | | | | |
| 18–30 years old | 18.3 | 16.0 | 22.6 | 19.8 | 0.003 |
| 30–55 years old | 54.9 | 58.8 | 66.4 | 62.1 | |
| >55 years old | 26.8 | 25.2 | 11.1 | 18.1 | |
| Gender | | | | | |
| Male | 26.8 | 15.4 | 15.7 | 17.5 | 0.08 |
| Female | 73.2 | 84.6 | 84.3 | 82.5 | |
| Education | | | | | |
| Primary | 5.8 | 37.4 | 52.1 | 39.7 | |
| High school | 43.5 | 39.8 | 36.6 | 38.8 | <0.001 |
| Tertiary | 50.7 | 22.2 | 11.3 | 21.5 | |
| Employment status | | | | | |
| Employed | 66.6 | 44.7 | 48.6 | 50.4 | |
| Unemployed | 13.0 | 23.1 | 36.3 | 28.2 | <0.001 |
| Homemaker | 8.7 | 14.6 | 6.1 | 9.2 | |
| Retired | 11.6 | 17.7 | 9.0 | 12.1 | |
| Transportation mode | | | | | |
| Walk | 11.3 | 41.7 | 67.2 | 49.8 | |
| Public transport | 15.5 | 23.5 | 26.0 | 23.5 | <0.001 |
| Private car | 73.2 | 34.8 | 6.8 | 26.7 | |
| Distance to supermarket (min) | | | | | |
| 0–10 | 88.7 | 52.7 | 37.9 | 51.1 | |
| 11–30 | 8.5 | 38.2 | 50.2 | 39.4 | |
| More than 30 | 2.8 | 9.2 | 11.9 | 9.5 | <0.0001 |
| Food security status | | | | | |
| Food secure | 57.7 | 36.7 | 30.1 | 40.0 | <0.001 |
| Food insecure | 42.3 | 53.0 | 69.9 | 60.0 | |
| **Shopping Characteristics** | | | | | |
| No of people shopped for: Mean (SD) | 3.68 (2.4) (%) | 4.48 (4.8) (%) | 3.91 (1.9) (%) | 4.05 (2.8) (%) | 0.84 |
| Shopping pattern | | | | | |
| Daily | 32.4 | 32.6 | 24.7 | 28.4 | <0.001 |
| Weekly | 60.6 | 41.7 | 45.2 | 46.7 | |
| Monthly | 7.0 | 25.8 | 30.1 | 24.9 | |
| Shopping for | | | | | |
| Self | 28.2 | 22.0 | 26.9 | 25.6 | 0.51 |
| Household | 71.8 | 78.0 | 73.1 | 74.4 | 0.60 |
| Main household shopper (Yes) | 62.0 | 83.3 | 81.7 | 78.9 | 0.01 |
| Responsible for food preparation (Yes) | 70.4 | 85.6 | 79.0 | 79.6 | 0.04 |
| Factors affecting supermarket choice | | | | | |
| Price | 35.2 | 50.0 | 48.9 | 46.9 | <0.001 |
| Convenience | 49.3 | 25.8 | 18.7 | 26.1 | |
| Value for money | 4.2 | 7.6 | 11.4 | 9.0 | |
| Quality | 4.2 | 4.5 | 7.8 | 6.2 | |
| Others | 7.1 | 12.1 | 13.2 | 11.8 | |
| Shopping area (outshopping) | | | | | |
| High SEA | 100 | 8.3 | 3.7 | 21.3 | <0.001 |
| Middle SEA | 0 | 87.9 | 20.0 | 37.9 | |
| Low SEA | 0 | 3.8 | 76.3 | 40.8 | |
| Main supermarket (Yes) | 85.9 | 78.8 | 88.6 | 85.1 | 0.05 |
| Shopping for promotions/sales (Yes) | 18.3 | 18.2 | 8.7 | 13.3 | 0.02 |

SEAs: Socioeconomic areas. Data is presented as proportions (%) based on indicated numerator (N), except when stated otherwise (e.g., Mean, (SD); *p*-values determined based on Chi-Square ($X^2$).

**Table 2.** Self-reported purchase frequency and preference of food categories classified by residential SEAs.

| Variables | Residential SEAs | | | |
|---|---|---|---|---|
| | **High SEAs** | **Middle SEAs** | **Low SEAs** | ***p*-Value** |
| N (%) | 71 (16.8) | 132 (31.3) | 219 (51.9) | |
| Frequency | % | % | % | |
| **Fruits and vegetables** | | | | |
| More than once a week | 41.1 | 40.0 | 33.1 | |
| Once a week | 42.2 | 43.1 | 29.1 | <0.001 |
| Once/twice a month | 16.7 | 16.9 | 37.8 | |
| **Meat** | | | | |
| More than once a week | 27.8 | 21.9 | 22.7 | |
| Once a week | 37.8 | 33.1 | 24.4 | 0.05 |
| Once/twice a month | 34.4 | 45.0 | 52.9 | |
| **Snacks** | | | | |
| More than once a week | 21.1 | 32.5 | 33.1 | |
| Once a week | 27.8 | 29.4 | 22.1 | 0.20 |
| Once/twice a month | 51.1 | 38.2 | 44.7 | |
| Purchased SSBs (Yes) | 66.2 | 55.3 | 62.1 | 0.30 |
| **Preferred bread type** | | | | |
| White | 28.2 | 36.6 | 20.5 | |
| Brown | 52.1 | 40.5 | 69.9 | <0.001 |
| Whole wheat | 15.5 | 13.0 | 3.7 | |
| No preference | 4.2 | 9.9 | 5.9 | |

SEA: Socioeconomic area; SSBs: Sugar-sweetened beverages; Data is presented as proportions (%) based on indicated numerator (N), except when stated otherwise (e.g., Mean, (SD)). *p*-values determined through Chi-Square ($X^2$) Tests.

### 3.3. Neighborhood Food Environment Perceptions

Factors associated with perceptions of the neighborhood food environment are presented in Table 3. When compared to respondents with a tertiary education, shoppers with a primary and high school education had 24% and 50% lower odds respectively of concurring that healthy foods were too expensive. In addition, those with primary education had 3.5 times higher odds of agreeing that the quality of fruits and vegetable in their environment was poor ($p < 0.05$) and approximately twice the higher odds of perceiving that supermarkets were not adequate ($p < 0.05$) in their neighborhood. The retired shoppers had approximately five times higher odds compared to the employed of being neutral on the affordability of healthy foods in their neighborhood. Conversely, low-SEA shoppers were less likely to be neutral compared to high-SEA shoppers in their perception that healthy foods are too expensive.

Shoppers who were food insecure and from low SEAs were less likely to consider healthy foods expensive. However, they were less likely to agree that fruits and vegetables were available in their local neighborhoods and had higher odds of considering the quality of fruits and vegetables in their neighborhoods to be poor.

### 3.4. Out-Shopping

Characteristics of out-shoppers and in-shoppers from the low and middle SEAs are presented in Table 4. Out-shoppers were 33% less likely to be unemployed and to walk to the supermarket in which they were intercepted, and they had 5.1 and 2.3 times higher odds of relying on public transport or private motor vehicles respectively compared to in-shoppers. Furthermore, out-shoppers were less likely to travel less than 30 min to shop compared to the in-shoppers.

**Table 3.** Adjusted multinomial regression results of the associations of individual-level factors with food environment perceptions.

| | Perceptions about Neighborhood Food Environment | | | | | | | | | | | | | | | |
| | *The Healthy Foods in Stores in My Neighborhood Are too Expensive* | | | | *It is Easy to Purchase Fruits and Vegetables in My Neighborhood* | | | | *The Quality of Fruits and Vegetables in My Neighborhood Is Poor* | | | | *There Are No Supermarkets in My Neighborhood* | | | |
| | OR | 95% CI | OR | 95% CI | OR | 95% CI | OR | 95% CI | OR | 95% CI | OR | 95% CI | OR | 95% CI | OR | 95% CI |
| **Variables** | Agree | | Neutral | | Agree | | Neutral | | Agree | | Neutral | | Agree | | Neutral | |
| Education (Tertiary) [a] | | | | | | | | | | | | | | | | |
| Primary level | **0.24 *** | **0.12, 0.45** | 0.54 | 0.24, 1.22 | 0.69 | 0.36, 1.30 | **1.81** | **0.66, 4.94** | 3.45 ** | 1.78, 6.47 | 1.50 | 0.73, 3.11 | 2.02 * | 1.08, 4.09 | 1.73 | 0.76, 3.92 |
| High school level | **0.50 *** | **0.28, 0.92** | 0.61 | 0.27, 1.39 | 0.91 | 0.48, 1.71 | **1.73** | **0.63, 4.75** | 1.59 | 0.86, 2.96 | 0.92 | 0.46, 1.84 | 1.59 | 0.83, 3.09 | 1.02 | 0.48, 2.32 |
| Employment (Employed) [b] | | | | | | | | | | | | | | | | |
| Retired | 0.16 | 0.91, 5.12 | **4.67 *** | **1.62, 13.44** | 0.01 | 0.43, 2.39 | 0.37 | 0.11, 1.26 | 0.71 | 0.31, 1.69 | 1.15 | 0.42, 3.12 | 0.64 | 0.28, 1.52 | 0.44 | 0.14, 1.44 |
| Unemployed | 0.78 | 0.48, 1.29 | 1.26 | 0.65, 2.45 | 0.35 | 0.80, 2.27 | 1.08 | 0.53, 2.21 | 0.72 | 0.43, 1.18 | 0.94 | 0.52, 1.72 | 1.27 | 0.77, 2.11 | 0.62 | 0.32, 1.21 |
| Residential area (High SEAs) [c] | | | | | | | | | | | | | | | | |
| Low SEAs | **0.04 *** | **0.02, 0.11** | **0.19 *** | **0.06, 0.63** | **0.35 *** | **0.15, 0.81** | **0.29 *** | **0.09, 0.90** | 6.36 *** | 2.69, 15.03 | 0.68 | 0.31, 1.51 | 1.69 | 0.80, 3.55 | 1.61 | 0.59, 4.40 |
| Middle SEAs | **0.11 *** | **0.04, 0.28** | 0.33 | 1.00, 1.14 | **0.42 *** | **0.18, 0.99** | 0.48 | 0.16, 1.49 | 3.42 ** | 1.45, 8.04 | 0.74 | 0.34, 1.61 | 0.86 | 0.40, 1.88 | 1.46 | 0.53, 3.96 |
| Food security (food Secure) [c] | | | | | | | | | | | | | | | | |
| Food insecure | **0.55 *** | **0.34, 0.87** | 0.68 | 0.37, 1.25 | 0.68 | 0.37, 1.25 | 0.72 | 0.44, 1.86 | 1.71 * | 1.05, 2.77 | 1.12 | 0.64, 1.95 | 0.91 | 0.56, 1.47 | 0.86 | 0.46, 1.59 |

Reference categories are in parenthesis. Variables of interest considered were: Education, employment, OR: Odd ratio was estimated at 95% confidence intervals (CI); ref: Reference. Perceptions about food and food environment were taken as affirmative (Agree). [a] adjusted for age, sex, and employment; [b] adjusted for age, sex and education; [c] adjusted for age, sex, employment and education. * $p$-values of <0.05, ** $p$-values of <0.001, *** $p$-values of <0.0001.

**Table 4.** Adjusted logistic regression results of the associations of individual-level factors of out-shopping vs. in-shopping in the low- and middle-socioeconomic areas.

| Outshopping | | |
|---|---|---|
| Variables | OR | 95% CI |
| [a] Education (ref: Tertiary level) | | |
| 　Primary level | 0.60 | 0.27, 1.33 |
| 　High school level | 0.70 | 0.31, 1.57 |
| [a] Employment (ref: Employed) | | |
| 　Retired | 0.27 | 0.17, 1.82 |
| 　Unemployed | 0.31 * | 0.24, 0.88 |
| [a] Transportation to shop (ref: Walk) | | |
| 　Private car | 2.16 * | 1.0, 4.68 |
| 　Public transport | 5.04 * | 2.64, 9.70 |
| [a] Distance to supermarket from home (ref: More than 30 min) | | |
| 　0–10 min | 0.12 * | 0.05, 0.28 |
| 　11–30 min | 0.26 * | 0.12, 055 |
| [a] Food security (ref: Food secure) | | |
| 　Food insecure | 0.85 | 0.49, 1.50 |
| [a] Food environment perceptions (ref: Disagree) | | |
| 　The healthy foods in stores in my neighborhood are too expensive | 0.77 | 0.41, 1.45 |
| 　It is easy to purchase fruits and vegetables in my neighborhood | 0.81 | 0.43, 1.53 |
| 　There are not enough supermarkets in my neighborhood | 1.33 | 0.73, 2.42 |
| 　The quality of fruits and vegetables in my neighborhood is poor | 3.05 *** | 1.53, 6.08 |
| [a] Self-reported frequency of food purchase (ref: Once-twice a month) | | |
| 　Fruits and vegetable | | |
| 　More than once a week | 2.30 * | 1.08, 4.90 |
| 　Once a week | 1.98 | 0.91, 4.27 |
| Snacks | | |
| 　More than once a week | 2.34 * | 1.18, 4.65 |
| 　Once a week | 3.16 * | 1.57, 6.36 |
| Meat | | |
| 　More than once a week | 1.32 | 0.62, 2.64 |
| 　Once a week | 1.40 | 0.75, 2.64 |
| [a] Purchased SSB (ref: no) | 0.94 | 0.54, 1.64 |
| [a] Factor influencing supermarket choice (ref: Convenience) | | |
| 　Price | 1.29 | 0.60, 2.74 |
| 　Value for money | 0.99 | 0.30, 3.07 |
| 　Quality | 3.19 * | 1.08, 9.40 |

[a] Adjusted for age and sex; OR: Odd ratio was estimated at 95% confidence intervals (CI); ref: Reference. Perceptions about food and food environment were taken as affirmative (Agree). * $p$-values of <0.05.

Furthermore, out-shoppers had approximately three times higher odds of perceiving the quality of the fruits and vegetables in their neighborhoods as poor. Similarly, they were also three times more likely to shop in supermarkets that they perceive sell quality food than shop because of convenience when compared to in-shoppers (Table 4).

There were significant differences between the groups in the self-reported purchase frequencies of snacks and fruits and vegetables. Out-shoppers reported having higher odds of purchasing snacks and fruits and vegetables more frequently compared to in-shoppers. However, the frequency of meat, and SSBs purchases were not significantly different between in-shoppers and out-shoppers.

## 4. Discussion

This study highlights the low and middle SEA shoppers' perceptions of the lack of available quality fruits and vegetables in their retail food environment. Fruits and vegetables were less frequently consumed in lower SEAs. Notwithstanding, the frequent purchase of snacks and SSBs is clearly evidenced in all the neighborhoods. The study also shows the high prevalence of food insecurity,

unemployment, and low level of education of shoppers from low SEAs. It further indicates that shoppers of lower educational status, food security and socioeconomic status did not perceive healthy food as expensive. It demonstrates differing perceptions of the neighborhood food environment between in-shoppers and out-shoppers, and because the latter generally have better access to transportation, they have access to a greater range of food choices.

### 4.1. Food Choice

Our data on consumption frequency patterns conform to findings that the socioeconomic status of one's residential location influences the consumption of healthy foods, especially fruits and vegetables. According to a study on food consumption done in an informal low-income area in Cape Town using a food frequency questionnaire and 24-h recall, only about a third of people consume fruits, and six out of every ten people consume vegetables on a daily basis [30]. Conversely, persons from high SEAs in our study, similar to reports by other studies in Africa, purchase meat more frequently than shoppers from middle and low SEAs [11,31,32]. In South Africa, there is a general partiality for meat. Persons from a higher socioeconomic group purchase meat slightly more frequently than persons of low socioeconomic status, as the poor often opt for cheaper, lower quality and less nutritious meat rather than not buying meat at all [33].

The high frequency purchase of SSBs and snacks by shoppers from all the socio-economic areas is in line with reports on their general consumption in South Africa [32]. A recent study conducted in similar economically disadvantaged communities of South Africa, had shown that more than a third of the study population had consumed 10 and more servings of SSBs per week (an average of one and half cans of soda of 330 mL per day) [34]. The consumption of SSBs and snacks has been attributed to be a major factor in the increasing weight gain worldwide, a fact which is evident in South Africa [35].

### 4.2. Neighborhood Food Environment Perceptions of Shoppers from High, Middle and Low Residential SEAs

Consumer perceptions regarding the availability, affordability and diversity of choices of nutritious food in their neighborhood food environment have been known to play a significant role in purchase and food choice, and consequently eating a balanced and nutritionally adequate diet [36,37]. The perceived lack of availability of quality fruits and vegetables in lower SEAs who are more food insecure in the current study was consistent with the results of a study that reported that supermarkets in low SEAs i. South Africa stock less healthy and lower quality foods compared to supermarkets in high SEAs [18]. Moreover, the quality, packaging and safety of fresh produce in less formal retail outlets in South Africa, such as the spazas, street vendors and convenience stores, which often located in low and middle SEAs, are often questionable [38]· Additionally, residents in low SEAs often have greater access to high-calorie and nutritionally poor foods, than fresh fruits and vegetables [3]. For example, the low consumption of fruits and vegetables in low-income areas from studies conducted in Seychelles and South Africa were attributed to financial constraints often leading to the substitution of the purchase of fruits and vegetables for cheaper high energy-dense, but low nutrient foods [5,31].

Contrary to reviews from both high- and low- and middle-income countries, a high percentage of shoppers from low SEAs, educational status and food security status indicated that healthy foods were not expensive. This is despite evidence that in fact, healthier foods are often more expensive, both in South Africa, and in other countries [3,39]. In our study, this is also confirmed by most of the shoppers from low and middle SEAs indicating that price was the main influential factor in their supermarket choice. Therefore, we can assume that one of the barriers to purchasing "healthy" foods for low-income individuals is cost. However, it is possible that the reason for low SEA shoppers' perception that healthy foods are not expensive is that the contradictory it is likely that there are different definitions of "healthy" food in low, middle and high SEAs in South Africa. For example, a study conducted in Kanana in Guguletu, a low-income area in Cape Town [11] showed that most of the population classified foods, such as maize meal (*Isidudu*), snacks, such as chicken crisps, sweets,

puffed corn, and imported SSBs, such as Fanta and Lemon Twist, as "healthy". The low level of education of low- and middle-socioeconomic persons in the study, may also be a contributory factor to health and nutrition knowledge. Education level influences food choices as people with increasing higher level of education are more aware of the nutritional quality of foods [4]. Further studies on the neighborhood food environment and the understanding of the definition of healthy foods are needed to better understand the challenges of access to healthier food, particularly in low-socioeconomic communities in South Africa. Such studies should include actual grocery receipts analysis or measures of expenditure, household ethnography and formal and informal retail environment surveys.

### 4.3. Outshopping

Our findings that out-shoppers were only from middle and low SEAs and predominantly from low-SEAs is in line with other studies in LMICs that indicate that individuals from low SEAs often shop outside their neighborhood [40,41]. However, the relatively small number of out-shoppers (12.0% and 23.7% middle and low SEAs respectively) in our study could be due to the increased presence of supermarkets and/or less formal retail stores in their communities and the lack of mobility of shoppers from the low SEAs. Previous studies in South Africa have found that most respondents from low SEAs walk to their shopping destinations, due to the high rates of unemployment, consequently being financially constrained and unlikely to own a car or able to afford frequent transport fare to shop [11,42]. The out-shoppers in this study were more likely to be employed and primarily used public transport compared to walking (49.3% vs. 31.3%) to get to their shopping location. It is therefore a possibility that the out-shoppers, as supported by a study from Soweto (a low-income neighborhood in Johannesburg, South Africa), were often employed outside their residential areas. Consequently, they tended to shop in proximity to their workplace after work or along their travel route and transportation hubs [9], where supermarkets are often strategically located for convenience [9,20]. Quality is also notably one of the key factors reported to influencing out-shopper's supermarket choice. As they perceive the quality of fruits and vegetables to be poor in their neighborhood, out-shoppers were more likely and able to travel outside their neighborhood for better quality produce compared to in-shoppers. Further, as there were no out-shoppers from high SEAs, it is most likely that the shoppers from high SEA also work in a high SEA. As shown in the study, they are satisfied with the number of supermarkets and quality of fruits and vegetables in their neighborhood and consequently have no reason to shop far from home, despite that most of them have private cars to transport their groceries if they shop outside their neighborhood. The findings in our study when assessing the high-, middle- and low-SEA shoppers is somewhat contrary to the findings in Soweto where high socioeconomic status, car ownership and a higher level of education were associated with outshopping [10]. However, looking into the subpopulation of in-shopper vs. out-shoppers within the low and middle SEAs, there are similarities with the findings in the Soweto study.

### 4.4. Demographics and Shopping Patterns of Shoppers from Low-, Middle- and High-Residential SEAs

Most of the respondents in this study from three different socioeconomic areas shopped in supermarkets, in agreement with studies conducted in low-income households in urban and peri-urban areas of Msunduzi municipality, KwaZulu-Natal province, South Africa [43,44]. The rapid expansion and distribution of supermarkets in South Africa appear to have greatly formalized shopping behavior, despite the increase of other less formal retail outlets, such as spaza shops, convenience stores and street vendors [19,45]. For instance, the number of supermarkets in Cape Town increased by 164% from 1994 to 2012 with the highest increase in low socio-economic areas [45]. Furthermore, the availability of more food varieties, both healthy and unhealthy, and lower prices are advantages that supermarkets have over spaza shops and convenience stores [43,44].

All the respondents from high SEAs, and most respondents from low and middle SEAs (>75%) do their major shopping in their neighborhood. Weekly shopping was a general norm in all the neighborhoods. Contributing to the cause of this shopping pattern for their major shopping in the

study might be the proximity of supermarkets to homes as most were in-shoppers. Lack of access to transportation and storage facilities are other factors that might contribute to the major weekly shopping pattern for shoppers residing in the low and middle SEAs. This is because walking might hinder them from purchasing more than they can carry. Additionally, many may not have adequate storage and preservation facilities, such as refrigerators to store fresh produce that will last them more than a week. This was recently confirmed by a study done in Langa, a low socio-economic area in Cape Town [46]. Approximately only a third of the population in Kanana, Gugulethu, another low SEAs in Cape Town has either refrigerator (31%) or a freezer (2%) [11]. Furthermore, according to South Africa Community Survey, 2016, approximately 20% of South African population, especially in the low SEAs, do not have a refrigerator.

As expected, shoppers in our study were predominantly women who were mostly responsible for household food decisions, including the purchase and preparation of food. This is a very typical scenario in an African setting where women primarily make decisions related to food consumed in the household [11,47]. The fact that the shoppers from high SEAs were significantly less likely to be responsible for food preparation, may reflect a gender bias, in that men were more well-represented in our intercept interviews in the high SEAs compared to those in the low and middle SEAs. It could also be due to the possibility that high-SEAs households employed more domestic help. Modernization, urbanization and influence of education are resulting in slight changes in the typical African settings, especially in the high income households, where the division of households and domestic chores, such as grocery shopping and food preparation are embraced jointly to an extent, or out-sourced even though women are still mostly responsible for food decision [48]. As anticipated, price was reported as an important factor influencing the supermarket choice of shoppers from low- and middle-SEAs as they seemingly have less disposable income and will shop where they will pay less for more. It is however surprising that even though price seems to be important in the supermarket choice of these socio-economic groups, sales/promotions did not influence their supermarket choice. This could be due to regular price reductions and promotions by most supermarkets for profit maximization [49].

## 5. Study Strengths and Limitations

The study captured the shopping characteristics of the high-, middle- and low-SEA urban South African as supermarkets have been shown to be the main retail food environment for most households. In addition, the study highlighted shopping characteristics and neighborhood food environment perceptions that vary socioeconomically which are key for intervention strategies to alleviate food insecurity and obesity.

The study only focused on shoppers in supermarkets and did not intercept shoppers from less formal food purchasing outlets, such as spazas and convenience stores. The frequency of purchase of the food types was self-reported and did not give details on the quantity and quality of the food bought. The understanding of what defined "healthy" foods was not assessed, and as such, may itself have varied according to SEAs, food security status and/or education. The measures (statements) for assessing the different dimensions of the neighborhood food environment used mostly one question to assess each dimension as opposed to multiple questions which can be summed, and the scores weighted thereby making them sometimes more reliable for statistical calculations. Height and weight of the participants were not measured and therefore, the body mass index (BMI) of the participants could not be assessed to provide context concerning body weight status (obesity) and food choices.

## 6. Conclusions

Food choice may be influenced by both neighborhood food environment and socioeconomic factors. Persons living in settings, such as low SEAs compared to high SEAs may only have access to foods of poor quality, with fewer choices, which is an indication of the differences in their food environment in the study setting. Notwithstanding, the frequent purchase of snacks and SSBs is clearly evidenced in all the neighborhoods. Sustainable intervention strategies to improve the quality of fruits

and vegetables in lower socioeconomic areas and reduce the consumption of snacks and SSBs in South Africa are paramount to reduce the prevalence of obesity and food insecurity.

**Author Contributions:** Conceptualization, E.V.L., N.L. and T.P.; methodology, E.V.L., N.L., T.P., L.M., F.O.-W., S.M. and K.O.; validation, M.U. and R.D.; formal analysis, F.O.-W., K.O. and O.A.; data curation, F.O.-W. and K.O.; supervision, F.O.-W., K.O. and H.V.; project administration, K.O. and E.V.L.; writing—original draft preparation, F.O.-W.; review and editing, R.D., O.A., L.M., S.M., N.L., H.V., L.T., J.B., T.P. and K.O.

**Funding:** DST-NRF Centre of Excellence for Food Security; URC University of Cape Town Post-Doctoral Fellowship.

**Acknowledgments:** Nandipha Sinyanya is thanked for an exceptional fieldwork support.

**Conflicts of Interest:** The authors declare no conflict of interest. The funders had no role in the design of the study; in the collection, analyses, or interpretation of data; in the writing of the manuscript, or in the decision to publish the results.

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
