# Peer review of "Food Purchasing Characteristics and Perceptions of Neighborhood Food Environment of South Africans Living in Low-, Middle- and High-Socioeconomic Neighborhoods"

_sustainability, doi:10.3390/su10124801_

Round 1

Reviewer 1 Report

The manuscript explores the purchasing behaviour in various areas in SA living in various socioeconomic communities and had attempted to link this data with health. The instrument that the authors had used is fine, but further analysis needs to be done to confirm their results.

The legend for Figure 1 looks cropped off for the High SEAs. Please fix.

Consider in making Table 1 more clearer, perhaps bold each categories (e.g age, gender) to make reading a bit more easier. 

Table 2. Spell out > Once

Table 3. Spacing adjustment is required, table looked messy (see Low 95%CI)

Validation of the scales and sample group is required in this study, consider running Cronbach Alpha, KMO analysis, and Factor Analysis to support the validation of the instrument.

I'm also not entirely convinced with the results from the logistic regression where the odd ratio is discussed. Logistic regression is used when we use binary response but it seems that the authors have performed data transformation on Section 2.6.6 from a 4 point scale and further on Section 2.7. Is there a reason why this was done?

Author Response

Dear Reviewer,

Thank you for your all your comments.  We have carefully examined and addressed each of the comments or concerns raised and have made changes as required or suggested in the  revised article with tracked changes. The reviewers did not suggest that the manuscript should undergo extensive English editing. However, all aspects of the manuscript  have been checked for grammar, and consistency.

Below are detailed responses to each of your valuable comments. We have referred to the sections in the revised manuscripts where changes, if required have been made. 

Point 1: The manuscript explores the purchasing behaviour in various areas in SA living in various socioeconomic communities and had attempted to link this data with health. The instrument that the authors had used is fine, but further analysis needs to be done to confirm their results.

Response 1:   We thank the reviewer for this comment and agree. For this purpose, we have undertaken additional research, which will help to confirm our findings, but is beyond the scope of this paper. This work is and will include in-depth interviews, neighborhood food environment audits, in a select, low-income cohort and will be extended to high-income areas.  So, at present, we can only report on the perceptions of the food environment, and the self-reported shopping behaviours or characteristics of persons shopping in high-, middle- and low-income areas.  

Point 2:The legend for Figure 1 looks cropped off for the High SEAs. Please fix.

Response 2:  Figure 1 has been deleted from the manuscript as it may not be suitable for modification and it is also not essential to the manuscript.

Point 3: Consider in making Table 1 clearer, perhaps bold each categories (e.g. age, gender) to make reading a bit more easier. 

Response 3: Each category in Table 1 (e.g.  age, gender) have been boldened.

 Point 4: Table 2. Spell out > Once

Response 4:In Table 2, >Once has now been written as More than once. 

Point 5: Table 3. Spacing adjustment is required, table looked messy (see Low 95%CI).

Response 5:Spacing adjustments in Table 3 have been done to improve readability. 

Point 6: Validation of the scales and sample group is required in this study, consider running Cronbach Alpha, KMO analysis, and Factor Analysis to support the validation of the instrument.

Response 6: We thank the reviewer for the suggestion to assess inter-item reliability test of the neighborhood food environment questions. However, each statement for measuring perceptions of  the neighbourhood food environment  assessed  a unique dimension.  These four statements were  adapted from a previous study which evaluated the neighborhood food environment in a low income neighbourhood in the United States[28]. One pertained to perceptions of the general neighborhood retail food environment, two were specific to fruits and vegetables (availability and quality) in the neighborhood  and  another to the affordability of healthy foods in the  neighborhood retail food environment (formal and less formal retail outlets). Representative statements included: ‘There are no supermarkets in my neighborhood ’, It is easy to purchase fruits and vegetables in my neighborhood’, ‘ The healthy foods in stores in my neighborhood are too expensive’ and ‘ The quality of fruits and vegetables in my neighborhood  is poor.’ Shoppers were asked these statements using a 5-item Likert scale coded 1-5 (1 strongly agree, 2 agree,3 neither agree or disagree, 4 disagree, 5 strongly disagree). The Likert scales (1-5) were subsequently collapsed into 3 categories and recoded as: strongly agree and agree=1 (agree); somewhat agree or disagree=2 (neutral); disagree and strongly disagree=3 (disagree).

 Point 7: I'm also not entirely convinced with the results from the logistic regression where the odd ratio is discussed. Logistic regression is used when we use binary response but it seems that the authors have performed data transformation on Section 2.6.6 from a 4 point scale and further on Section 2.7. Is there a reason why this was done?

Response 7:  We thank the reviewer for the accurate observation. Multinomial regression analysis was done and results are presented in Table 3, not logistic regression as  previously indicated. All analyses have been rechecked for accuracy. In Table 3, agree, neutral and disagree responses were analysed, only the agree vs disagree outcomes were previously shown but the neutral responses have now been included in the Table 3 and checked for accuracy. 

 We appreciate the time taken to review the article.

Reviewer 2 Report

The authors present an interesting study about characteristics and factors that influence food purchase decisions of residents of Low, Mid or High income neighbourhoods in Cape Town, South Africa.  As the authors also highlight in the introduction, there is limited work in this area from lower and middle-income countries. Moreover, the authors specifically look at SES differences within low/mid income countries. Therefore, the study nicely contributes to the scientific literature. However, some questions raised during the review process, especially on the completeness of the description of the measurements and statistical analyses used and outcomes of Table 4. Therefore, major revisions are needed prior to publication.

TITLE

Please adjust the title, as I would expect a different study… e.g. include something on “evaluation of food purchase characteristics in supermarkets in low, mid, high income neighbourhood”.  Neither the retail food environment nor purchase patterns have been examined.

ABSTRACT

Please specify the main research question + the measures used.  

Line 24” can you be a bit more specific about the ‘ factors associated with food purchasing decisions”?” there are many factors related. What did you study specifically?

Line 27-29: compared to who were these shoppers more likely?

 It is stated in the abstract that lack of mobility is associated is stated as key issue for low SEA shoppers. However, I cannot read this in the presented results in the abstract.  

INTRODUCTION

Please add reference for line 40-41

Line 51-52: “the effect” of supermarket choices and food purchasing decisions… can the authors specify the effect “on what”   

52 – What do the authors mention with food shopping patterns? – And do you actually measuring patterns? Of is the study on food purchase decisions?

52-53: This sentence introduces already the aim of this paper (and it is framed that no studies have been conducted yet. It would be of interest to clarify why we need this study, what will these insights bring us?  So, why is this of interest?   I think the rest of the paragraph introduces this to a certain amount already, and I think you could delete this sentence. /

In line with this, In-shoppers and out-shoppers + transport mode are central in the paper/results. The introduction would benefit from highlighting the need for the inclusion of these determinants in scientific research somewhat more comprehensively. Why do the authors expect this would be different?

At the end of the introduction, an overall aim is presented, however it would be beneficial to specifically formulate the aim(s) of this study.

METHOD

What is an intercept questionnaire? I am not familiar with this terminology.  And the authors conduct an interview or an oral survey? In section 2.4 it is an interview, and in 2.6 an survey. As the data is analysed in a quantitative manner, suggest not using the word interview? As I would expect a qualitative analyses.

Line 106: were only supermarkets from one supermarket chain included? or different chains included? Please clarify.

Line 111: shopped at ONE of the identified supermarkets?

 Line 132-132: please rephrase. Not clear what ‘sub-population with unique shopping characteristic’ is. + if possible, would you also have looked at the out-shoppers that had a high SEP, if yes, please mention here that this was the case for all participants. In the result section you can frame that this was not the case for high SEP participants.    

Line 146: do you mean NUMBER of persons shopped for?

Line 146:  What do you mean by “persons responsible for shopping/preparation? If this was the responsibility of the shopper or someone else of the household?

Line 146: did you use an existing survey to determine ‘factors that influence choice of supermarket?”

147-148: what do you mean by ‘major shopping destination”?

Par 2.6.2. How were “the food types” and “frequency of purchases “measured? E.g. what measure was used? Was this measure validated? Idem for section 2.6.3. How were these preferences measured? And were these preferences in line with the products measured in section 2.6.2?

Par 2.6.4. This paragraph should obtain a different title: e.g. perceptions on the availability of healthy food in people’s neighbourhood. Also for this section, did you used existing measures?  Furthermore, What was the survey question? Was the statement “healthy foods are expensive” specifically questioned for food available in one’s neighbourhood, or for foods in general?

Line 163; gender = sex   

How was ‘educational attainment’ measured? How was ‘employment status’ measured? How was ‘SEA’ measured? Alternatively, was the latter based on zip code? Please clarify.  

Par 2.7 data analyses.  This paragraph would benefit from a bit more details. Moreover, I think if the research question/aim of the study is framed more clearly at the beginning, the conducted analyse make more sense.

Nevertheless, how was logistic regression analyses conducted with the “food environment perception” as dependent variable,  while this variable had no 2 but 3 outcome options (agree, neutral, disagree)  . Ordinal/Categorical regression analyses needed.  Please specify as remains unclear for reader.

Also here, alter the variable name “ food environment perception “)

In the abstract, also, differences in perceptions for in-shoppers and out-shoppers are presented, but this have not been described clearly in the analyses section. Please add.

line 187: add “IBM“ SPSS?

RESULTS

Line 186: Delete “gender” (sex is already mentioned)

196-198: With respect to the outcomes on commuting mode (striking outcomes), is there also data on the distance of the supermarket-home? E.g. to make sure it is not the case that individuals with a higher SEP need to travel a larger distance.

You mention the most used commuting modes. Hower, I find it also an interesting finding that only 6.8% of individuals with lower SEP go by car oppose to 73.3% of individuals with higher SEP. Only 11.3% of individuals with higher Sep walk (oppose to 67.2 of individuals with lower SEP). You might want to highlight this as well.

Line 200-203: would these differences influenced the findings? Alternatively, did you adjust for this in the analyses?

Line 218: Do the authors mean that low-SEA shoppers PREFERRED brown bread more? Instead of PURCHASED?

Paragraph 3.3. – line 228. Please check the used analyses and the outcomes in table 3.

TABLE 1:

Please add “participants” in the title of this Table

Please add the total N, top right

TABLE 2

Why was only the preference of bread-type questioned? In addition, why was it questioned if participants (Yes/No) purchased SSB in another way than for fruit/veg/meat/snacks?

Please clarify in the method sections why these choices were made.

 In line 154, it is mentioned that SSB preference is measured, but I cannot find this in the Table.

TABLE 3:

Please use the same variable names as presented in the method section in line 160-161

Again, I do not understand how a logistic regression was conducted using categorical outcome variables (preference variables).

OR if food insecurity and fruits and Veg not need to be bold as not stat sig.

The ORs of the outcomes by poor quality of fruits and vegetables are incorrect: some lie not within the 95%-CI.. bv. An OR of 0.40, cannot have an 95%ci of 1.79-6.47… please check the analyses.

TABLE 4:

Wat was the rational for presenting the unadjusted outcomes?

The authors tested if out-shopping was related to different food-consumption purposes. What was the rational for studying this? Why do you expect a difference for in-shoppers. Would be of interest to read the rationale / hypothesis in the introduction

DISCUSSION

Please reframe the first paragraph with results once analyse have been re-done. Please provide the main results of the main research questions of this study.

Line 295-298: I don’t find the bread-argument very strong for the fruit/vegetable difference in purchases. It only argues that bread is already cheaper, irrespective of the zero-tax. It does not say anything on SEP differences (and contradicts the results presented in line 312-313).

Line 312-317: I do not completely follow the reasoning (315-317) for the opposite finding in this study. Could you please clarify?

Line 333-335: so it could be the case they shopped after work (in a high SES area) prior to going home again? And is it the case that individuals with a high SES are likely to work and live in a high SES area?

Line 372-375: The result section is lacking results on sales/price promotions although this is mentioned here as outcome.. Moreover, a reason could be as well that not all food choices are made well deliberately (People like to think that sales/promotions do not steer their behaviour). So might be the case as well.  

Section 5 of discussion. I miss the strengths of the study.

Not sure if needed for this journal, but recommendations for further research/practice/policy are currently not in the discussion (limited mentioned in conclusion). Check if this should be added.

CONCLUSION

Please define clear conclusions based on the presented study.

Line 385 –delete “in conclusion”

First time the term ‘ecological’ is used… change or introduce earlier in the manuscript

Line 386-387:  statements that are not based on the study results of this paper

Author Response

Dear Reviewer,

Thank you for your all your comments.  We have carefully examined and addressed each of the comments or concerns raised and have made changes as required or suggested in the  revised article with tracked changes. The reviewers did not suggest that the manuscript should undergo extensive English editing. However, all aspects of the manuscript  have been checked for grammar, and consistency.

Attached are detailed responses to each of your valuable comments. We have referred to the sections in the revised manuscripts where changes, if required have been made. 

Thank you.

Reviewer 3 Report

Interesting work involving the proper data and the analysis, however  I suggest some minor changes:

1) SSB - There is the SSB explanation in line 225, however, the best alternative would be to explain SSB in the abstract (line 32) and in the main text (line 217, page 6) in order to make it clearer for the reader.

2) There is a small mistake in the number of table (page 7) - change it into 2. please.

3) Check precisely the table no 1. (percentages and the number of consumers; you have put the number N=422 in the abstract and the main text, however there is N=419 in total here; there is also e.g. mistake in Food security status in the  Middle SEAs part).

Author Response

Dear Reviewer,

Thank you for your all your comments.  We have carefully examined and addressed each of the comments or concerns raised and have made changes as required or suggested in the  revised article with tracked changes. The reviewers did not suggest that the manuscript should undergo extensive English editing. However, all aspects of the manuscript  have been checked for grammar, and consistency.

Below are detailed responses to each of your valuable comments. We have referred to the sections in the revised manuscripts where changes, if required have been made. 

Point 1: SSB - There is the SSB explanation in line 225, however, the best alternative would be to explain SSB in the abstract (line 32) and in the main text (line 217, page 6) in order to make it clearer for the reader.

Response 1:The full meaning of the acronym  “SSBs” (sugar sweetened beverages) has been included as suggested in Line 36 and 249 of the revised manuscript.

Point 2: There is a small mistake in the number of table (page 7) - change it into 2. please.

Response 2: All the tables have been checked and correctly numbered. The second Table has been relabeled Table 2.

Point 3: Check precisely the table no 1. (percentages and the number of consumers; you have put the number N=422 in the abstract and the main text, however there is N=419 in total here; there is also e.g. mistake in Food security status in the Middle SEAs part).

Response 3: In Table 1,the overall total number is supposed to be (N=422). There was an error in the total number of shoppers from low and middle socioeconomic areas. The correct number are N=219 for low SEA and N= 132 for middle SEA. The percentages and the number for shoppers from high SEA remains unchanged. The correct number of respondents for each SEA has been corrected in the manuscript.

The percentage (37.2%) for food secure middle SEA has been changed to the correct percentage 47% after reanalyzing.

Thank you once again for the time taken to review the article.

Round 2

Reviewer 1 Report

I'd like to thank the author for addressing all the comments.

For my previous point on Point 7 in regards to the regression, please include more information in the data analysis section why multinomial regression has been chosen in this study. 

However, it is important to note that in the future it is important for the authors to have multiple questions to measure one dimension here rather than having one singular question for one dimension. Please add this as well in the weakness section before the conclusion.

Author Response

Dear Reviewer,

Thank you for your comments.  We have carefully examined and addressed each of the comments or concerns raised and have made changes as required or suggested in the revised article with tracked changes. The reviewers did not suggest that the manuscript should undergo extensive English editing. However, all aspects of the manuscript  have been checked for grammar, and consistency.

Below are detailed responses your comments. We have referred to the sections in the revised manuscripts where changes, if required have been made. 

Point 1: For my previous point on Point 7 in regards  to the regression, please include more information in the data analysis section why multinomial regression has been chosen in this study. 

Response 1: Multinomial regression is often used to predict nominal dependent variables with more than two categories for one or more independent variables. We tested for multicollinearity between the independent variables and found no substantial issues as no variation inflation factor (VIF) exceeded 3. This has been included in the  data analysis section 2.7

Point 2: However, it is important to note that in the future it is important for the authors to have multiple questions to measure one dimension here rather than having one singular question for one dimension. Please add this as well in the weakness section before the conclusion.

Response 2: The following sentence has been included in the section 5( Strengths and Limitations) of the manuscript as a limitation “The measures (statements) for assessing the different dimensions of the neighbourhood food environment used mostly a single question to assess each dimension as opposed to multiple questions which can be summed, and the scores weighted thereby making them sometimes more reliable for statistical calculations.

Thank you.

Reviewer 2 Report

The authors addressed the review in detail and made the requested changes to this interesting paper. Thank you

Author Response

 Dear Reviewer,

Thank you for the time taken to review the manuscript.